# Antibacterial Activity and Mechanism of GO/Cu_2_O/ZnO Coating on Ultrafine Glass Fiber

**DOI:** 10.3390/nano12111857

**Published:** 2022-05-29

**Authors:** Manna Li, Zhaofeng Chen, Lixia Yang, Jiayu Li, Jiang Xu, Chao Chen, Qiong Wu, Mengmeng Yang, Tianlong Liu

**Affiliations:** 1International Laboratory for Insulation and Energy Efficiency Materials, College of Materials Science and Technology, Nanjing University of Aeronautics and Astronautics, Nanjing 211106, China; nuaa_lmn@163.com (M.L.); ljy0822@mail.nwpu.edu.cn (J.L.); xujiang73@nuaa.edu.cn (J.X.); chenchaojsw@163.com (C.C.); wuqiong@nuaa.edu.cn (Q.W.); yangmengmeng@nuaa.edu.cn (M.Y.); liu1061178707@nuaa.edu.cn (T.L.); 2Jiangsu Collaborative Innovation Center for Advanced Inorganic Function Composites, Nanjing University of Aeronautics and Astronautics, Nanjing 211106, China; 3Suqian Kongtian New Materials Co., Ltd., Suqian 223800, China

**Keywords:** GO/ZnO/Cu_2_O antibacterial powder, antibacterial coating, antibacterial properties, ultrafine glass fibers, antibacterial mechanism, *E. coli* and *S. aureus*

## Abstract

A GO (graphene oxide)/ZnO/Cu_2_O antibacterial coating was successfully sprayed on the ultrafine glass fibers using room temperature hydrothermal synthesis and air spraying techniques. The microstructures of the antibacterial coating were characterized, and the results showed that the Cu_2_ONPs (nano particles)/ZnONPs were uniformly dispersed on the surface of GO. Then, the antibacterial properties of the GO/ZnO/Cu_2_O (GZC) antibacterial coating were evaluated using the disc diffusion test. It was found that the coating exhibits excellent antibacterial properties and stability against *E. coli* and *S. aureus*, and the antibacterial rate of each group of antibacterial powder against Staphylococcus aureus (*S. aureus*) and Escherichia coli (*E. coli*) was 100%. To explore the antibacterial mechanism of the GZC antibacterial powder on the ultrafine glass fibers based on the photocatalysis/oxidative stress method, the photoelectric coupling synergistic effect between GZC antibacterial coating was analyzed deeply. The results all showed that the photochemical activity of GZC antibacterial powder was significantly improved compared with pure component materials. The enhancement of its photochemical activity is beneficial to the generation of ROS (including hydroxyl radicals, superoxide anion radicals, etc.), which further confirms the speculation of the photocatalytic/oxidative stress mechanism.

## 1. Introduction

Metal nanoparticles have attracted major research value due to their outstanding electronic, optical and microstructure [1,2,3,4,5,6,7,8]. GO is a new type of material that is filled with multiple layers of carbon atoms to form a two-dimensional lattice [9,10,11,12,13,14]. GO has potential utilization in various applications such as renewable energies, electronic devices, transistors, sensors and solar cells [15]. GO contains a large number of oxygen-containing functional groups such as hydroxyl, carboxyl and epoxy bonds and can be divided into single layer or multilayer [16,17,18]. GO has an important platform for the distribution of metal nanoparticles, related to its high surface area and the large synthetic stability consequences between GO and metal nanoparticles [19,20]. In recent years, research on the antibacterial properties of graphene have gradually increased. In contrast, there is little mention of the sterilization and inactivation of graphene loaded with a variety of metal oxides [21].

Many metal (silver–Ag, gold–Au, palladium–Pd) and metal oxide (zinc oxide ZnO, titanium dioxide TiO_2_, copper oxide CuO, cuprous oxide Cu_2_O, aluminum oxide Al_2_O_3_, magnesium oxide MgO) nanomaterials have attracted the attention of researchers due to their antibacterial properties [22,23,24,25,26,27]. Among them, ZnO has a high light absorption rate in the long-wave ultraviolet (315–400 nm) and ultraviolet (280–315 nm) regions, showing good antibacterial properties. In addition, ZnO has FDA-approved bio-safety and is used as an effective antibacterial drug in the medical field due to its bio-compatibility [28,29,30,31]. Chang et al. synthesized an inexpensive and readily available ZnO/CuO nanocomposite by a two-step hydrothermal method [32]. Hafsa Siddiqui et al. investigated the biosynthesis and photocatalytic and antibacterial activities of flower-shaped copper oxide nanostructures [33]. Amna Sirelkhatim et al. studied the antibacterial activity and toxicity mechanism of ZnO nanoparticles [29]. Cu_2_O is one of the most widely used bactericidal reagents at present. The main reason is that the method of preparing Cu_2_O is very simple and the raw material price is relatively low. Secondly, the bactericidal and inactivation effect of Cu_2_O is really excellent. The principle of Cu_2_O sterilization and inactivation is that Cu_2_O can release a large number of copper ions to eliminate germs, and at the same time, it will also produce a large number of reactive oxygen species, which also has certain help and promotion for sterilization and inactivation [34]. Based on these points, it is particularly important to make the distribution of cuprous oxide nanoparticles more uniform and broader and to make the release of copper ions more effective in the period. 

Based on the above theories and studies, GO can be used not only to disperse nanoparticles of metal oxides as their support, but also to prolong the release of nanoparticles from metal oxides. In addition, due to a synergistic effect between graphene and metal oxide nanoparticles, the aging performance of metal particles can be greatly improved. In this study, a certain amount of ZnONPs and Cu_2_ONPs were loaded on the surface of graphene, and the antibacterial performance of the composite antibacterial powder could be further improved due to the synergistic effect of the three. Of course, both ZnONPs and Cu_2_ONPs are chemically treated and then loaded onto the graphene oxide sheets [21]. The obtained GZC antibacterial powder was prepared into an aqueous slurry and sprayed on the surface of ultrafine glass fiber, and, finally, antibacterial ultrafine glass fiber was prepared. The phase and micro-structure of the synthesized GZC antibacterial powder was characterized by X-ray diffraction (XRD), Fourier infrared absorption spectra (FT-IR), Scanning Electron Microscopy (SEM), high resolution transmission electron microscopy (HTEM) and other characterization methods. In addition, the antibacterial properties of metal oxide nanoparticles have also been studied and tested to a certain extent [21].

## 2. Experimental Details

A GZC antibacterial coating was deposited on the ultrafine glass fiber using room temperature liquid phase with air room temperature spraying technology, which includes three steps, as follows. Firstly, a simple room temperature liquid phase synthesis method was used to prepare GZC antibacterial powder, and a mixture of nano GO (40%, 30%, 20%), ZnONPs, CuSO_4·_5H_2_O and sodium dodecyl sulfonate (SDS) was ultrasonically dispersed. At room temperature, 0.075 g of copper sulfate pentahydrate was first placed in a conical flask, followed by 0.03 g of GO and spherical ZnONPs of the same mass, and, finally, 3.85 g of sodium dodecyl sulfonate. Weigh 300 g of deionized water and add it into a conical flask for ultrasonic dispersion, which lasted for 1 h. After that, the rotor was added for magnetic stirring for 3 h. After working overtime, 10 mL 0.2 mol/L ascorbic acid solution was added, then the pH value of the mixed solution adjusted to 9 by dropping 1 mol/L sodium hydroxide solution; mixing was continued for another 2 h. Agitation stopped as precipitation gradually appeared in the flask and the mixture was centrifuged. The obtained sediments were washed four to five times in deionized water and then dried at 80 °C to obtain GO/ZnO/Cu_2_O antibacterial powder [22]. Since the GO surface has oxygen-containing functional groups with adsorption and ZnONPs provide Cu^2+^ protection area, after adding the reducing agent ascorbic acid and sodium hydroxide solution to react, ZnONPs and the reduced Cu_2_ONPs can be uniformly dispersed on the surface of GO. The antibacterial powders obtained with 40%, 30% and 20% GO were named GZC-1, GZC-2 and GZC-3. Secondly, a certain amount of distilled water was added to the antibacterial powder (due to the presence of surfactants in the antibacterial powder) and ultrasonically dispersed for 10 min to prepare 1%, 3% and 5% water-based slurries; the acrylic resin and water were mixed according to the ratio of 3:1 the mass ratio and ultrasonically dispersed for 10 min, and kept for later use. Finally, the room temperature spraying method was used to spray the polyacrylic resin solution on the surface of the ultrafine glass fiber, and the aqueous slurry was then sprayed and air-dried at 110 °C for 15 min. The process is schematically illustrated in Figure 1. Through electrostatic interaction and condensation reaction, the antibacterial water-based slurry was evenly coated on the surface. An antibacterial coating with a controllable thickness was formed on the surface of the ultrafine glass fiber to exert a long-term inhibitory effect on germs.

Meanwhile, Cu_2_ONPs and ZnONPs were also uniformly distributed and insulated by GO in the forming process of GZC antibacterial powder. Thus, the surface of 2D GO, containing many functional groups such as hydroxyl, carboxyl, epoxy and GO, has electrostatic adsorption capacity, and might effectively impede the agglomeration of the Cu_2_ONPs and ZnONPs during the preparation process. In order to compare the influence of different GO content on the antibacterial properties of antibacterial powders, the identical powders with different mass fractions of GO were synthesized with all other conditions that remained the same.

## 3. Results and Discussion

### 3.1. Morphological Analysis of GO/ZnO/Cu_2_O Antibacterial Powder

Figure 2a,b shows typical surface SEM morphology of the GZC antibacterial powders. It is plicated and needle-like. The average thickness of the pure GO nanosheets (NSs), based on SEM analysis, is 0.6–1.2 nm (Appendix A). Appendix A shows the morphologies of ZnO, which are analysed by SEM; its diameter is 20–30 nm and most of the ZnONPs are spherical. The acicular substance in the antibacterial powder may be Cu_2_O, which is 30–50 nm, suggesting the successful preparation of acicular cuprous oxide on the surfaces of GO NSs. The antibacterial powder contains 2D GO NSs, spherical ZnO, needle-like Cu_2_O in the antibacterial powder and different antibacterial effects will be shown, achieving antibacterial synergy and improving antibacterial efficiency.

The HRTEM pictures are displayed in Figure 2c–e. In the figures, the size of Cu_2_ONPs is less than 10 nm, which is relatively small, and the content is also less. The size of ZnONPs hardly changes, still being 20–30 nm. Both of them appear to be uniformly distributed on the surface of the GO sheet without agglomeration, and the agglomerated part may be residual surfactant and Na_2_SO_4_ that could not be removed after the reaction. The images of ZnONPs and Cu_2_ONPs lattice fringes are 0.281 nm and 0.213 nm, respectively, as shown in Figure 2d,e.

Figure 2f shows the EDS diagram of the GZC antibacterial powders. The figure reflects that the element types in the antibacterial powder are mutually confirmed with other test results. Among them, the copper content is low, which proves that the synthesized Cu_2_ONPs is less, and the presence of sulfur and sodium element indicates that the antibacterial powder contains residual surfactants. As shown in the Figure 2g, characteristic peaks of GO appear at 285 eV, but the partial peaks of Zn 2p and Cu 2p move back from 1021 eV and 931 eV to 1070 eV and 978 eV. It may be that the oxygen-containing functional groups in GO have a certain influence on the two substances.

### 3.2. Structural and Optical Analysis of GO/ZnO/Cu_2_O Antibacterial Powder

Figure 3a shows the XRD pattern of the antibacterial powder. The sample shows the main diffraction peaks are positioned at 2θ = 31.8°, 34.4°, 47.5°and 56.6°, which are assigned to (1 0 0), (0 0 2), (1 0 2) and (1 1 0) sttice planes of ZnONPs phase based on JCPDS 36-1451 [32]. The pattern indicates that Cu_2_ONPs were also formed in the antibacterial powder. The small and weak diffraction peaks are positioned at 2θ = 62.4°and 74.4°, which are assigned to (2 2 0) and (3 1 1) sttice planes of Cu_2_ONPs phase based on JCPDS 05-0667 [33]. The intensity of the diffraction peak corresponding to the characteristic pattern of Cu_2_O is relatively weak, indicating that the content of Cu_2_ONPs in the prepared antibacterial powder is relatively low. This is also verified in Figure 2e. The (0 0 1) diffraction peak from GO [35] is almost invisible, since its structure changes after the metal oxide is compounded on the surface of GO. For the XRD pattern of antibacterial powder with sharper diffraction peaks of Na_2_SO_4_, its main diffraction peaks are positioned at 2θ = 19.0°, 23.1°, 28.0°, 28.9°and 38.6°, that may be derived from the surfactant sodium dodecyl sulfonate (C_12_H_25_SO_3_Na) used in the preparation of antibacterial powders.

Figure 3b shows the FT-IR spectrum of GZC antibacterial powders and the separated pure components Cu_2_O, ZnO and GO nanopowders. The FT-IR spectrum shows that GO has absorption peaks at 3420 cm^−1^, 1721 cm^−1^ and 1060 cm^−1^, corresponding to hydroxyl (-OH), carbonyl (-C=O) and hepoxy. The stretching vibration of the base is (-C-O-C) [36]. ZnO has a standard characteristic absorption peak at 510 cm^−1^, and Cu_2_O also has a typical Cu-O stretching vibration peak at 631 cm^−1^ [37]. Since the Cu_2_O powder used in the test was precipitated by the liquid phase reduction method, the shape of the FT-IR spectrum is very similar to the GZC antibacterial powder; the absorption peak at 3420~3450 cm^−1^ corresponds to the stretching vibration of the hydroxyl group, and the adsorbed water molecules on the surface are accounted for. The two high-intensity absorption peaks at 2922 cm^−1^ and 2852 cm^−1^ correspond to the stretching vibration of the C-H bond, and the C-H bond at 1470 cm^−1^ the flexural vibration absorption peak may be derived from the organic surfactant used in the preparation of the powder sample. The absorption peaks at 1650 and 1060 cm^−1^ correspond to the stretching vibrations of -C=O and -C-O-C. They all originate from the presence of GO in the composite, and due to variations in the composite of materials, some characteristic absorption peaks of ZnO and GO show a tendency to shift to high wave values in the FT-IR spectrum of composite materials. Onal groups were introduced, indicating that the performance of the sample remained stable during the test without deterioration or oxidation.

### 3.3. Antibacterial Properties

In this study, the antibacterial properties of powders were tested by the plate test and the disc diffusion test. The specific test methods were carried out in accordance with the GB/T 21510-2008 “antibacterial properties of nano-inorganic materials”. Firstly, a certain amount of bacterial strains was activated. In order to obtain the growth curve of bacteria, spectrophotometer method was used in this study to draw, and then the growth cycle of activated bacteria was obtained, and relatively stable strains were selected as the test of antibacterial performance in the experiment. For the strains used in this study, Luria-bertani medium (LB medium) was selected as the substrate for bacterial culture. The formula for LB medium and additional phosphate buffer solution required is roughly as follows: first, add 1 g of sodium chloride, then, 1 g of peptone, then add 0.5 g of yeast extract and 1 mol/L of sodium hydroxide, for a total of 4 mL. The phosphate buffer consists of 0.8 g of sodium chloride, 0.02 g of potassium chloride, 0.027 g of potassium dihydrogen phosphate and 0.355 g of disodium phosphate dodecahydrate. The formulas refer to the international standard “Method for Testing Antibacterial Properties of Inorganic Materials (GB/T 21510-2008)”. The above sodium hydroxide solution was added to LB medium and phosphate buffer solution, respectively, and their pH values were adjusted to 7.2–7.4. Before using LB medium and phosphate buffer solution, they should be placed at 121 °C for sterilization and inactivation for at least 20 min.

The antibacterial performance test method in this study was as follows: first, LB medium was poured into the petri dish and cooled sufficiently before being used. Then, phosphate buffer was used to dilute the activated bacterial solution from 10^9^ cfu/mL to 10^5^ cfu/mL. Next, 0.01 g of the above prepared GZC antibacterial powder and 1 mL of the newly prepared bacterial solution were accurately weighed. The two compositions were shaken fully in the test tube to evenly mix them, and then let stand for 12 h at a constant temperature. At the same time, 0.01 g of control sample was accurately weighed for the same treatment as just performed. After that, the upper layer of the bacteria liquid of each sample was evenly coated on the surface of the culture medium, and then the culture dish was placed in a constant temperature incubator at 37 °C for 24 h. Finally, the culture dish was taken out, and the bacteriostasis rate of each group was calculated by the colony counting method. The bacteriostasis rate *R* was calculated by:(1)$$R=\frac{A-B}{A}\times 100\%$$
where *R* stands for antibacterial rate, %; *A* is the number of colonies in the control group, per; *B* is the number of bacterial colonies in experimental group, per.

Then came the operation process of the bacteriostatic circle experiment. First, a piece of filter paper was cut in a circular shape, and some samples were added to deionized water to form a 1 mg/mL solution. The circular filter paper was immersed in the solution and then dried. A part of activated *E. coli* was taken and spread on a petri dish covered with AGAR and then placed in a biochemical incubator with the temperature set at 37 °C for 24 h. After being taken out, the size of the bacteriostatic zone was checked, and then the treated circular filter paper was placed on the filter paper, and the bacteriostatic zone was measured again [38].

The plate test is used to study the inhibitory effect of GZC antibacterial powder on *E. coli* and *S. aureus*, and silica powder is used in the control group. The antibacterial powder was mixed with the test bacteria solution, which was applied to the plate after continuous contact for 12 h and then incubated at a constant temperature for 24 h. The colony growth of the two bacteria is shown in Figure 4. The results of the inhibition rate test are recorded as follows in Table 1.

From Figure 4a–c and Figure 5a–c, it can be seen that after the GZC antibacterial powder with different proportions was in contact with the bacterial solution for 12 h, no colonies were found in the culture medium. However, a large number of colonies grew in the medium using silica powder as a control. Figure 4d,h shows that the colonies of *S. aureus* and *E. coli* are about 630 and 210, respectively. Therefore, the antibacterial rate of each group of antibacterial powder against *S. aureus* and *E. coli* is 100%. GZC antibacterial powder has excellent antibacterial activity against *S. aureus* and *E. coli*.

According to Section 2. 18 of the “Technical Specifications for Disinfection” promulgated by the Ministry of Health of the People’s Republic of China (the section on the experimental standard procedure of the antibacterial experiment), the bacteria to be tested are selected as *S. aureus* and *E. coli*, and the disc diffusion test is used to evaluate the antibacterial properties of the samples. The specific operation steps for performance are illustrated as follows: (1) Cut the sample to be tested according to the standard, then place it on the ultra-clean workbench and irradiate the sample with the ultraviolet lamp for later use. (2) Use a triangular spreader to dip the bacterial solution with a concentration of 10^6^~10^7^ cfu/mL after the gradient dilution, and evenly spread the bacterial solution on the surface of the medium plate, then cover the petri dish and place it at room temperature in an ultra-clean workbench dried for 5 min. (3) Place the sample to be tested gently on the surface of the nutrient agar medium plate, making sure that the samples are separated by a certain distance (>25 mm), and ensure that the sample is tightly attached to the surface of the plate. Finally, at a constant temperature of 37 °C, incubate in an incubator for more than 12 h. During this process, in order to prevent moisture from dripping onto the petri dish, keep it upside down. (4) Check and determine the diameter of the inhibition zone. The measurement range is based on the periphery of the inhibition zone, and the experiment needs to be repeated three times to improve the accuracy of the antibacterial performance of the test sample.

The antibacterial experiment was carried out with *E. coli* and *S. aureus* as the tested strain. The results are shown in Figure 5a–c and Figure 6a–c, by which it can be seen that after 24 h of constant temperature incubation, the GZC antibacterial coating samples of each group also showed obvious antibacterial activity against *E. coli* and *S. aureus*. It can be clearly seen that with the gradual increase in the content of antibacterial powder in the water-based coating (the mass fractions are 1%, 3% and 5%, in sequence), the diameter of the inhibition zone produced by the corresponding sample has also significantly expanded. Therefore, it can be preliminarily proved that the antibacterial coating can exhibit relatively excellent antibacterial activity and sensitivity to the two tested bacterial species.

Longitudinal comparison of the diameter of the inhibition zone of the GZC antibacterial coating on the two tested bacterial species can also be found; in the case of the same incubation time, the antibacterial coating is antibacterial against *S. aureus*. The diameter of the circle is larger than that of *E. coli*, that is, the antibacterial coating has stronger inhibitory effect on *S. aureus* than *E. coli*.

**Figure 6 nanomaterials-12-01857-f006:**
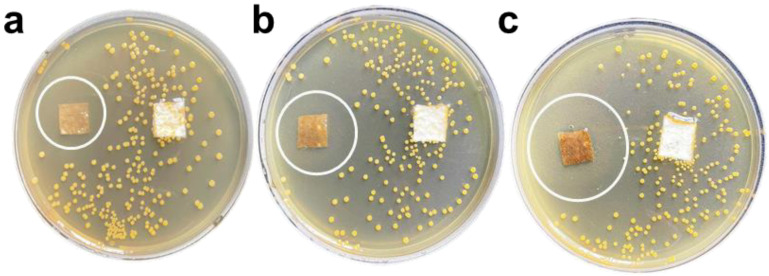
Antibacterial zone diagram of antibacterial coating on *S. aureus*, the mass fractions GZC antibacterial powder are 1% (**a**), 3% (**b**) and 5% (**c**), in sequence.

### 3.4. Photocatalysis/Oxidative Stress Mechanism

In this study, photoelectrochemical testing methods were used to analyze and explore the antibacterial mechanism of GZC antibacterial powder, and the specific manifestations of the synergistic and synergistic antibacterial effects between the various components of the composite material were examined. Specifically, it involves testing the light absorption wavelength range of each component material, light response properties, the number of active oxygen radicals generated under light, and the charge transfer and transfer efficiency.

ESR (Electron Spin Resonance) is a spectroscopic technique that can be used to track unpaired electrons in free radicals. It is the most direct and effective technical means to detect and study free radicals. In this study, the electron paramagnetic resonance spectrometer (Bruker A300, Brock Technology Co., Ltd, Mc Donald, PA, USA) was used, mainly using one of the commonly used oxygen radical traps; 5,5-dimethyl-1-pyrroline-N-oxide (DMPO) and ROS react to form more stable free radicals (called spin trapping), and the characteristic ESR signal of 1:2:2:1 was generated during the detection process [39].

The ESR method was used to determine the formation of O_2_^−^,·OH and ^1^O_2_, three oxygen free radicals in active oxygen. Among them, DMPO reacts with O_2_^−^,·OH to form [DMPO-O_2_^−^]^·^ adducts and [DMPO-OH]^·^ adducts, and TEMP reacts with ^1^O_2_ to form TEMPO. Pure GO and better GZC antibacterial powder (GO content 40%) was selected for the ESR test, and the ESR of GZC antibacterial powder that produces three active oxygen ESR signals under light and dark conditions was recorded. The spectrum is shown in Figure 7a–c, and the ESR spectrum of three active oxygen ESR signals produced by GZC antibacterial powder and pure GO under light conditions is shown in Figure 7d–f.

Figure 7a–c shows that the GZC antibacterial powder has obvious ESR characteristic peaks for the three active oxygen radicals under light conditions, but hardly produces characteristic ESR signals under dark conditions—that is, almost under dark conditions. No active oxygen free radicals are produced, indicating that light is a necessary condition for activating a large amount of ROS production.

Figure 7d–f shows that GO and GZC-1 antibacterial powders can generate a considerable amount of active oxygen radicals under light conditions, including O_2_^−^, ^1^O_2_ and ·OH, and GZC antibacterial powders with GO content of 40%; that is, the ESR signal intensities of the three active oxygen radicals produced by the powder are higher than pure GO to varying degrees. Figure 7d shows the formation of superoxide anion radicals. It can be clearly seen that the ESR signal intensity of [DMPO-O_2_^−^]^·^ produced in GZC-1 is higher than that of GO, indicating that due to the presence of ZnO and Cu_2_O, GZC-1 composite material can produce significantly more O_2_^−^ than pure GO under light. Figure 7e shows the formation of hydroxyl radicals; it can be clearly seen that the ESR signal of GO produced under light [DMPO-OH]^·^ is weak, and the ESR signal of [DMPO-OH]^·^ produced in GZC-1 is very strong under the same conditions, which shows that, compared with pure component GO, GZC-1 composite material produces ^1^O_2_ amount under light. It has been significantly improved; Figure 7f shows the formation of singlet oxygen. It can be seen that the ESR signal intensity of TEMPO produced by GZC-1 under light is also significantly higher than pure GO. In summary, GZC-1 produces more O_2_^−^,·OH and ^1^O_2_ than GO; that is, GZC-1 produces more active oxygen than GO under the same light conditions. Obviously, from the perspective of ROS sterilization, the GZC antibacterial powder with ZnO and Cu_2_O will show stronger antibacterial activity.

**Figure 7 nanomaterials-12-01857-f007:**
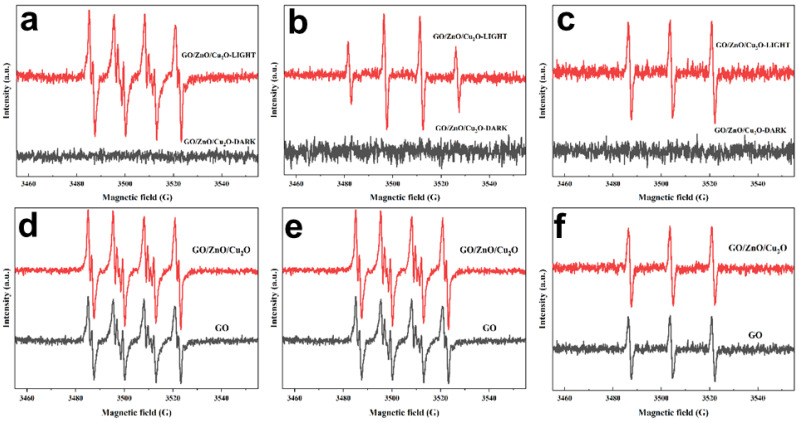
GZC antibacterial powder and GZC antibacterial powder and GO ESR spectra of three oxygen radicals produced under light and dark conditions. (**a****,d**) [DMPO-O_2_^−^] addition product. (**b****,e**) [DMPO-OH]^·^ addition product. (**c****,f**) TEMPO.

In order to further explore the effect of GO as a carrier on the light response properties of GZC antibacterial powder, this paper uses ultraviolet-visible spectrophotometry (UV-Vis) to analyze GZC antibacterial powders with different GO contents. Figure 8 shows the UV-Vis spectra of the GZC (GO content 40%) composite material and the pure substances of each component.

As shown in Figure 8, GO has an ultraviolet absorption peak at 230 nm, which is due to the π–π transitional absorption of the C=C bond on the GO aromatic ring [40]. ZnO has strong absorption in the ultraviolet region, and the edge of the absorption band is cut off at about 380 nm. Compared with ZnO and GO, the absorption wavelength range of GZC antibacterial powder shows a significant blue shift, that is, GZC antibacterial powder has a wider light absorption range than ZnO and GO. Secondly, by observing the UV-Vis spectrum, it can also be found that the spectral response range of GZC antibacterial powder moves to the visible light region compared to the single component. It can be considered that the presence of GO and Cu_2_O is beneficial in improving the photocatalytic performance of GZC antibacterial powder and promotes the transition of free electrons and improves the separation efficiency of hole–electron pairs, thereby generating more electrons and holes for the synthesis of active oxygen and improving the overall antibacterial properties of the antibacterial powder active. In addition, the use of the Kubelka–Munk function to draw the Tauc diagram can also estimate the band gap energy *E*_g_ of each sample material. The formula is as follows:(2)$${\left(\alpha hv\right)}^{1/m}=B(hv-E_{g})$$

In the Equation (2), *α* represents the absorption coefficient, *hν* can be approximately substituted into the calculation for discrete photon energy (Planck constant *h* = 6.62606896 × 10^−34^ J·s [41]), *B* is a constant and the value of m is determined by the specific semiconductor. The type of optical transition is determined (direct transition, *m* = 1/2; indirect transition, *m* = 2). The materials used in this article are all direct transition semiconductors, and the value of *m* is 1/2.

**Figure 8 nanomaterials-12-01857-f008:**
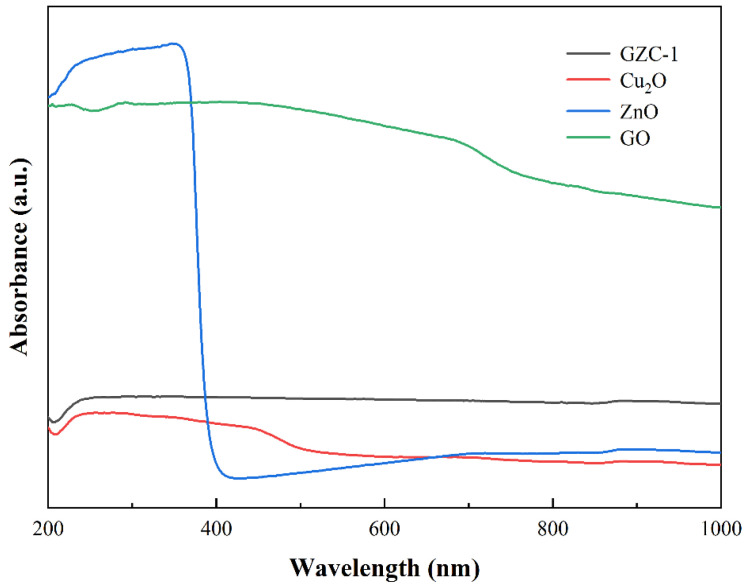
UV-Vis spectra of GZC antibacterial powder and pure substances of each component.

With (*αhν*) 2 and *hν* as the *x* and *y* axes, draw the Tauc diagrams of ZnO, Cu_2_O, GO and GZC antibacterial powder, respectively, as shown in Figure 9a–d, below.

**Figure 9 nanomaterials-12-01857-f009:**
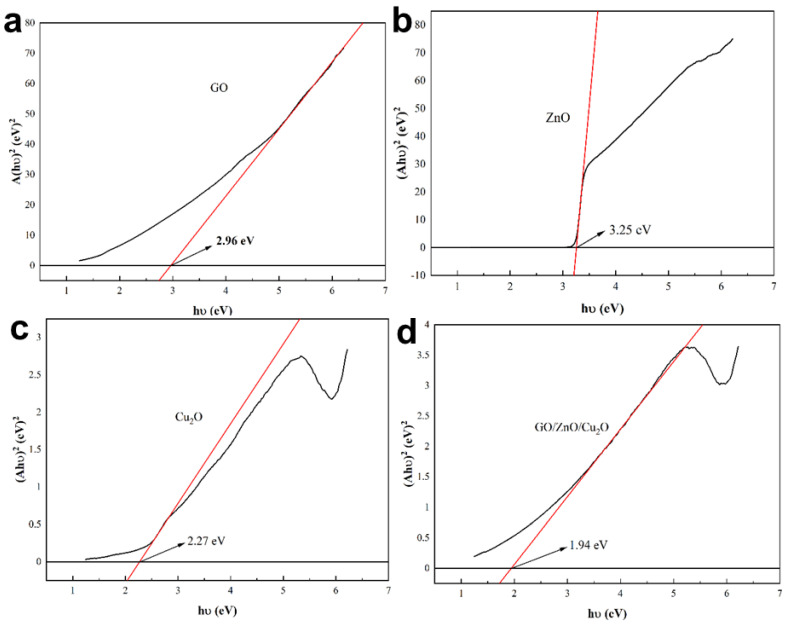
Tauc diagram of each powder sample. (**a**) GO. (**b**) ZnO. (**c**) Cu_2_O. (**d**) GZC antibacterial powder.

It can be seen from Figure 9a–d that the *E*_g_ of ZnO, GO, Cu_2_O and GZC antibacterial powder is about 3.25 eV, 2.96 eV, 2.27 eV and 1.94 eV. It also shows that there is a certain synergistic effect among ZnO, Cu_2_O and GO, which makes the band gap energy of GZC antibacterial powder lower than that of the pure substances of each component. According to this result, it can be qualitatively considered that the GZC antibacterial powder is more prone to electron excitation and transition under light conditions than the pure substances of each component. From another perspective, the improvement in the performance of the antibacterial powder in the light-induced sterilization mechanism is confirmed.

Electrochemical Impedance Spectroscopy (EIS), also known as AC impedance, is an electrochemical measurement method that uses a small-amplitude sine wave AC potential (or current) as an input disturbance signal. The system can be used to study the migration and transfer of photogenerated electrons and holes in photoinduced materials, including characterizing the kinetic mechanism of charge transfer reactions at the electrode/electrolyte interface and determining the electrical conductivity and charge transfer properties of electrodes. The result can be represented by a complex variable function with angular frequency *ω* as a variable, denoted as:(3)$$R=\frac{A-B}{A}\times 100\%$$
(4)$$\varphi =\arg\;tg\left(–Z\prime \prime /Z\prime \right)\;$$
where *Z* is the impedance and *ϕ* is the phase angle.

In this paper, the photocurrent response spectrum and EIS impedance spectrum are mainly used to qualitatively analyze the separation and transfer efficiency of photogenerated charges. With the stronger photocurrent response spectrum signal, the charge separation efficiency of the material is higher; the EIS impedance spectrum can be qualitatively divided into a low and high frequency region. The overall shape of its curve is “semi-circular arc + tail” type. Since the high frequency/low resistance region is mainly dominated by the charge transfer resistance; in general, the smaller the semicircular arc radius in the high frequency region, that is, the lower the electrochemical transfer resistance of the material, the higher the charge transfer efficiency of the material.

The photocurrent response spectra and EIS impedance spectra of the GZC antibacterial powders and the separated GO, ZnO and Cu_2_O powders are shown in Figure 10 and Figure 11.

It can be seen from Figure 10 that the photocurrent response intensity of GO/ZnO/Cu_2_O antibacterial powder is significantly stronger than that of other pure component materials, followed by ZnO. The results indicate that there is a certain synergistic effect after the combination of the three semiconductor materials, which can reduce the recombination of photo-generated electron–hole pairs in narrow-band gap materials such as Cu_2_O and GO, and at the same time promote the separation of photocharges in ZnO, which makes the new material have strong charge separation efficiency. According to Figure 11, the Nyquist plots of all samples in the high frequency/low resistance region show different degrees of semicircular arcs. Each semicircular arc in this region can represent a charge transfer process, where a larger arc radius corresponds to a larger transfer resistance [42]. An EIS system is used to study the migration and transfer of photogenerated electron–hole in photoinduced materials [43]. Generally speaking, the smaller the arc radius, the higher the charge transfer efficiency. It can be observed from Figure 11 that the semicircle radius of GZC antibacterial powder is smaller than that of other components, indicating that the addition of ZnO and Cu_2_O makes the GZC antibacterial powder have faster interfacial charge transfer and higher electron hole pair separation efficiency, thus further improving the germicidal effect of GZC antibacterial powder [44]. Observing the impedance curves of the four materials, it can be clearly distinguished that the semi-circular arc curves of pure GO and GZC antibacterial powders in the high frequency region are the most “gentle”, so it can be determined that they have smaller arcs. Compared with other pure component materials (ZnO, Cu_2_O), it can be qualitatively considered that GO and GZC antibacterial powder have higher charge migration and transfer efficiency. According to this conclusion, it can also be inferred that when the three materials are recombined, their internal structures have undergone some changes due to the recombination, and it can be reasonably inferred that GO and Cu_2_O can assist the photogenerated electron–hole pairs of ZnO to realize the interface charge through a certain process. The migration and transfer of the antibacterial powder significantly improved the antibacterial ability of the antibacterial powder from the perspective of photo-induced sterilization.

The detection results of UV-Vis spectrum, photocurrent response and EIS impedance spectrum all show that the photochemical activity of GZC antibacterial powder has been significantly improved compared with pure component materials. The enhancement of its photochemical activity is beneficial to the generation of ROS, which further confirms the speculation of the photocatalytic/oxidative stress mechanism. The antibacterial synergistic effect between antibacterial powders can be explained from the perspective of photoelectrochemical principles: GO and Cu_2_O, as semiconductor materials, can also generate photogenerated electron–hole pairs under illumination conditions. There are separation energy levels of metallic copper ions, and there may be chemical bonds when the powder materials are composited. These factors can promote the interface charge migration of ZnO during the photoexcitation process and improve the separation rate of photogenerated electron–hole pairs, resulting in more ROS being used to kill bacteria. The antibacterial mechanism is shown in Figure 12.

## 4. Conclusions

In this paper, the GZC antibacterial powder was synthesized by the room temperature liquid phase method, and the antibacterial water-based coating was prepared according to the concentration gradient. The coating was sprayed on the surface of ultrafine glass fiber cotton to form antibacterial coatings with different concentrations. The structure of the antibacterial powder was characterized by XRD (Neo-confucianism Company, Tokyo, Japan), SEM (Hitachi Instruments LTD., Tokyo, Japan), HRTEM (Hitachi Instruments LTD., Tokyo, Japan), and FTIR (Thermo Fisher Scientific Co., LTD, Waltham, MA, USA). Taking *S. aureus* and *E. coli* as the research objects of the antibacterial properties test, the antibacterial powder and antibacterial coating were tested by the plate test and inhibition zone method.

(1)The HRTEM images of the GZC antibacterial powder show that the two-dimensional lamellar structure of graphene oxide can support ZnO and Cu_2_O particles relatively uniformly, and its loose and porous structure can play a good role in antibacterial and slow release.(2)FTIR showed that after GO was compounded with ZnO and Cu_2_O, the structure of each component did not change significantly, and no new functional groups were introduced.(3)*S. aureus* reached 100% inhibition rate after exposure for 12 h; with the increase in the antibacterial coating concentration, its inhibition zone against *E. coli* and *S. aureus* gradually expanded, and the antibacterial effect on *S. aureus* increased better than *E. coli*. The application feasibility of the antibacterial coating was preliminarily verified.(4)This research focuses on the antibacterial mechanism of GZC antibacterial powder, puts forward hypotheses based on existing data, uses relevant test methods in the field of photoelectrochemistry to verify and analyze, and, finally, summarizes the photocatalysis/oxidative stress mechanism.

## Figures and Tables

**Figure 1 nanomaterials-12-01857-f001:**
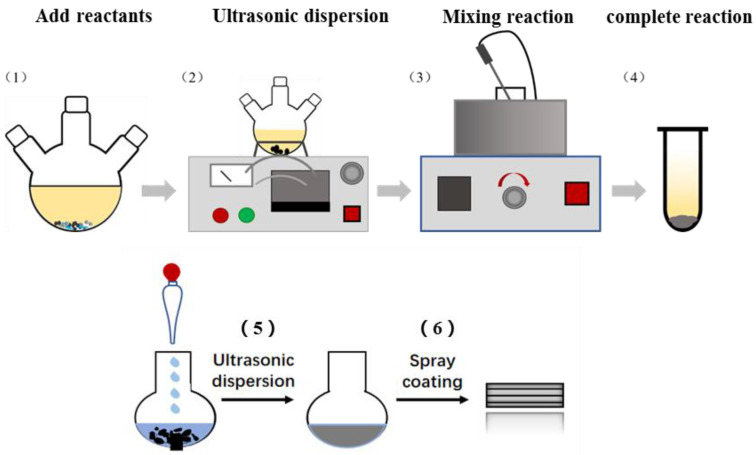
Process schematic diagram of the preparation procedures of the GZC antibacterial coating on the ultrafine glass fiber.

**Figure 2 nanomaterials-12-01857-f002:**
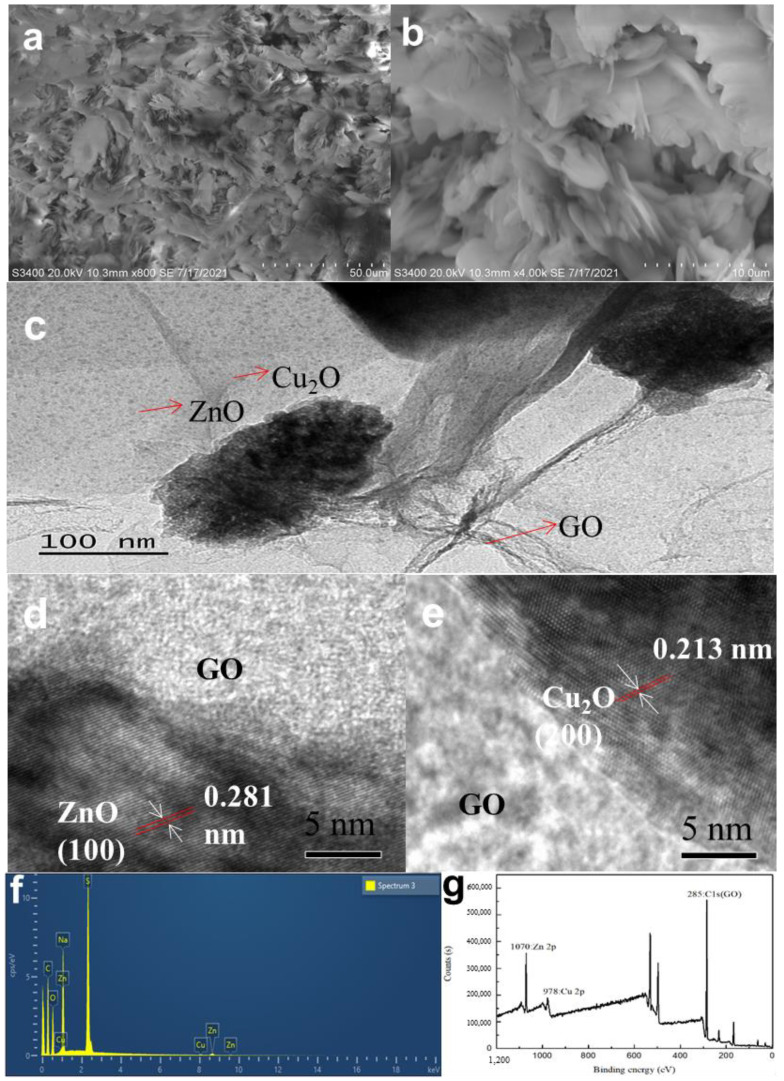
(**a,b**) SEM pictures. (**c**–**e**) HRTEM image. (**f**) EDS spectra. (**g**) XPS spectra of GZC antibacterial powder.

**Figure 3 nanomaterials-12-01857-f003:**
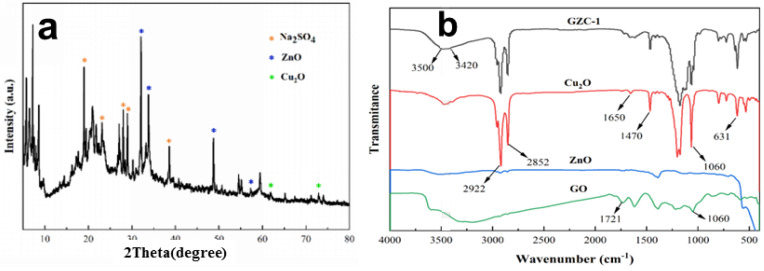
(**a**) XRD pattern. (**b**) FT–IR spectrum of GZC antibacterial powder.

**Figure 4 nanomaterials-12-01857-f004:**
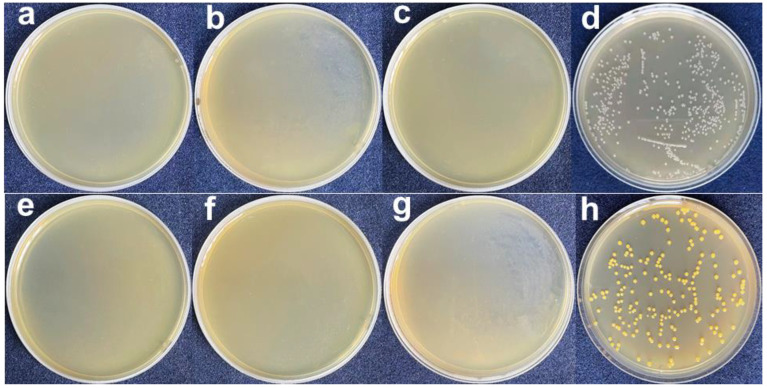
The picture of antibacterial powder to treat colony growth in *E. coli* and *S. aureus* (**a,e**) GZC-3. (**b,f**) GZC-2. (**c,g**) GZC-1. (**d,h**) Silica powder control.

**Figure 5 nanomaterials-12-01857-f005:**
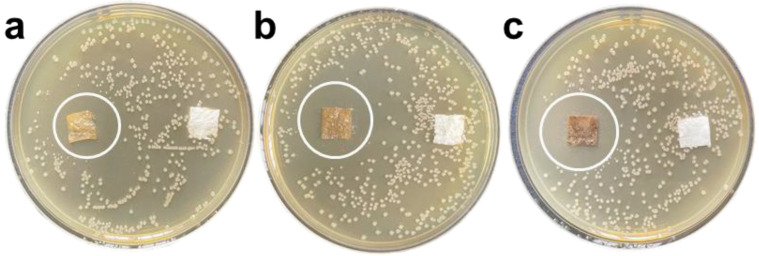
Antibacterial zone diagram of composite antibacterial coating on *E. coli*; the mass fractions of GZC antibacterial powder are 1% (**a**), 3% (**b**), and 5% (**c**) in sequence.

**Figure 10 nanomaterials-12-01857-f010:**
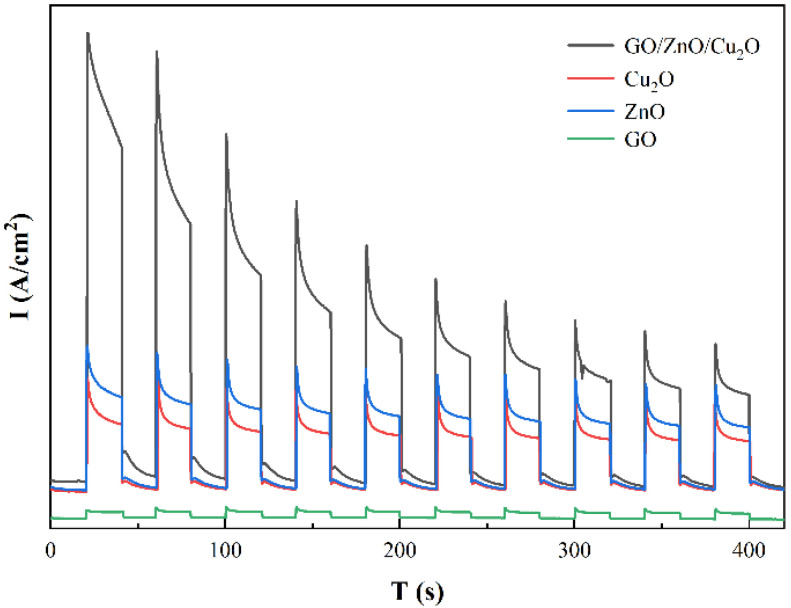
Photocurrent response spectra of GO, ZnO, Cu_2_O and GZC antibacterial powder.

**Figure 11 nanomaterials-12-01857-f011:**
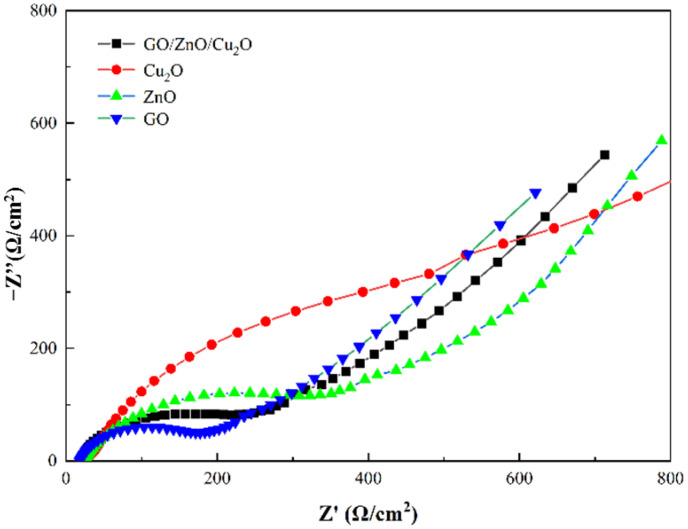
EIS impedance spectra of GO, ZnO, Cu_2_O and GZC antibacterial powder.

**Figure 12 nanomaterials-12-01857-f012:**
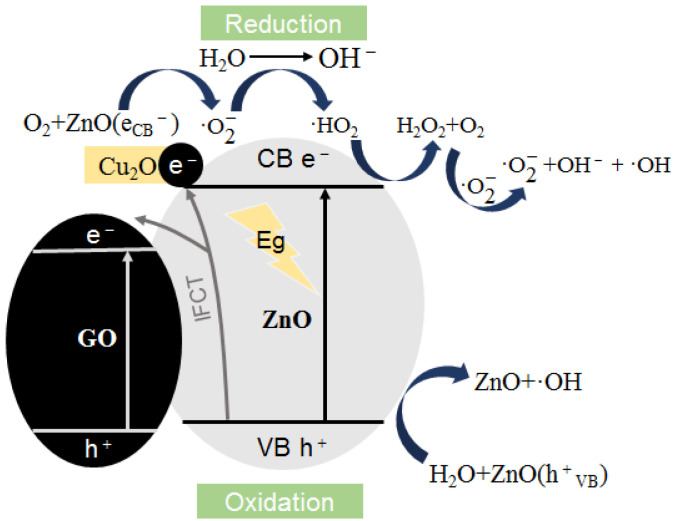
Schematic diagram of the mechanism of ROS formation.

**Table 1 nanomaterials-12-01857-t001:** Antibacterial rate of antibacterial powder against *E. coli* and *S. aureus*.

	Sample Serial Number	Clump Count/Indivual	Antibacterial Ratio/%
*E. coli*	GZC-1	0	100
	GZC-2	0	100
	GZC-3	0	100
	Control group	630	/
*S. aureus*	GZC-1	0	100
	GZC-2	0	100
	GZC-3	0	100
	Control group	210	/

## Data Availability

Exclude this statement if the study did not report any data.

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
