# Peer review of "Antibacterial Activity and Mechanism of GO/Cu2O/ZnO Coating on Ultrafine Glass Fiber"

_nanomaterials, 2022, doi:10.3390/nano12111857_

Round 1

Reviewer 1 Report

  1. It is recommended that multiple references be avoided in the introduction and later. All of these must be either deciphered or maximized in 2 references.
  2. I miss the fact that no abbreviation for GO/ZnO/Cu2O has been introduced. This long name should therefore be written unduly often throughout the entire duration of the article. For example, GZC or similar may be good, but this should then be applied everywhere, not only in the experimental part and graphs but also in the full text.
  3. In the introduction, I consider it unnecessary to indicate the complete and chemical marking of compounds (e.g. silver Ag). This article may appear in a scientific journal, its readers should be aware of the chemical signs.
  4. In Figure 1, the authors schematically present the synthesis, where the steps are marked from 1 to 4, and then highlight a part to 4-5. In addition, I miss the numerical reference to these steps in the text section, which would give you a better understanding.
  5. In Figure 2, electron microscopic images are marked from a to e  , and then f and g are missing.
  6. In Figure 3, in the XRD recording, "Thete" is defective, which is correctly Theta. Also here in the figure showing the results of FT-IR, there is a gray bar, the interpretation of which I could not find in the text. Could  this have been a picture error?
  7. Since 100% antibacterial results were detected in all samples, did it not occur to you that the active substances were used in excessive concentrations? Has there been a study on the lowest concentration of the composition?
  8. Figures 4 and 5 would be combined and presented in one.  The same applies to Figures 8 and 9.
  9. In all figures, the authors correctly use the same colors to present the results of each pattern, however, in Figure 13, ZnO and GO  are replaced in this respect. Please check this and improve it.

Author Response

Dear editor,

Thank you very much for your positive response. We have revised the manuscript (Nanomaterials-1738779) according to the comments and suggestions of the reviewers, and responded point by point to the comments as listed below.

We would like to resubmit this revised manuscript to Acta Acustica united with Acustica.

Yours Sincerely

Responses to the reviewers' comments:

First of all, we thank the reviewers for their comments and suggestions. The corresponding changes in the revised manuscript are marked in red color.

Reviewer 2 Report

  1. (silver Ag, gold Au, palladium Pd), it should write like ( silver-Ag, gold-Au, palladium-Pd).
    2. GO can be used not only to disperse nano-particles of metal oxides as their support but also to prolong the release of nanoparticles from metal oxides. What is this meaning needs more ellaborate discussions? 
    3. "Figure 1. Here if the authors can mention more details and include more steps of figures for a better understanding of readers if possible.
    For more confirmation about materials, the authors should add XPS results. 
    4. The captions of Figure 4. 5, and all others should rewrite properly. there are no commas, brackets, or demarcations marks. like a. or a).....
    5."Figure 5. The picture of antibacterial powder to treat colony growth in S. aureus a GO content 20%, b GO content 30%, c GO content 40%, d Silica powder control." Using GO only? or something mistake?
     "Table 1. " representations are confusing, GZC-1, GZC-2, GZC-3? not mentioned in text or caption,
    6. "Figure 6. and 7, Antibacterial zone diagram of composite antibacterial coating on E. coli, the mass fractions are 1% a, 3% b, and 5% c in sequence." Those captions are not proper descriptions .."composite antibacterial coating on E. coli? Not mentioned about coating substances.
    7. In EIS results, GO composites showed higher resistance than GO only. Why? 
    8. The title should change, the current title is very focused on antibacterials activity, needed to add other activities also and cover all manuscript areas. 

Author Response

(The authors gave the same response as above.)

Round 2

Reviewer 1 Report

The authors have made efforts to correct the article based on the corrections I have requested. Now I find the article suitable for publication in the journal of Nanomaterials.

Author Response

Dear editor,

Thank you very much for your positive response. We have revised the manuscript (Nanomaterials-1738779) according to the comments and suggestions of the reviewers, and responded point by point to the comments as listed below.

We would like to resubmit this revised manuscript to Acta Acustica united with Acustica.

Yours Sincerely

Responses to the reviewers' second comments:

First of all, we thank the reviewers for their comments and suggestions. The corresponding changes in the revised manuscript are marked in red color.

Reviewer 2 Report

Fig. 2 "g XPS spectra of GZC antibacterial powder", It should be like" XPS survey spectrum of ................................powder". Please denote every peak, Also possible to show Cu and Zn peaks. For reference see some literature. 

"XPS result has been added as Fig. 2g. As shown in the F igure, characteristic peaks of GO appear at 285 s, but the partial peaks of Zn and Cu may move backward and forward due to the combination, or may not appear due to the small content." Authors give a confusing reply, please should be defined correctly, shifted or not? appear or not, why? 

Figures 5, and 6 are not properly discussed in the caption or in the text. Must define what is the other treated materials in the plates. 

"7. In EIS results, GO composites showed higher resistance than GO only. Why? Answer: Thank you very much for your comment. The probable reason may be that when the three materials are combined, their internal structure changes due to the combination, and the GO and Cu2O can assist the photoelectron-hole pair of ZnO to achieve the transfer and transfer of interface charge through a certain way, which significantly improves the antibacterial ability of the composite powder from the perspective of photosterilization."  need references and should include in the text. 

Author Response

(The authors gave the same response as above.)
